# Potential Excess Intravenous Antibiotic Therapy in the Setting of Gram-Negative Bacteremia

**DOI:** 10.3390/pharmacy9030133

**Published:** 2021-08-03

**Authors:** Ashley R. Selby, Jaffar Raza, Duong Nguyen, Ronald G. Hall 2nd

**Affiliations:** Department of Pharmacy Practice, Jerry H. Hodge School of Pharmacy, Texas Tech University Health Sciences Center, Dallas, TX 75235, USA; Ashley.selby@ttuhsc.edu (A.R.S.); Jaffar.raza@ttuhsc.edu (J.R.); duong.nguyen@ttuhsc.edu (D.N.)

**Keywords:** antibiotic, anti-bacterial agent, antimicrobial, Gram-negative, duration, bacteremia, stewardship, patient discharge

## Abstract

(1) Background: Excessive intravenous therapy (EIV) is associated with negative consequences, but guidelines are unclear about when switching to oral therapy is appropriate. (2) Methods: This cohort included patients aged ≥18 years receiving ≥48 h of antimicrobial therapy for bacteremia due to *Escherichia coli*, *Pseudomonas aeruginosa*, Enterobacter, Klebsiella, Acinetobacter, or *Stenotrophomonas maltophilia* from 1/01/2008–8/31/2011. Patients with a polymicrobial infection or recurrent bacteremia were excluded. Potential EIV (PEIV) was defined as days of intravenous antibiotic therapy beyond having a normal WBC count for 24 h and being afebrile for 48 h until discharge or death. (3) Results: Sixty-nine percent of patients had PEIV. Patients who received PEIV were more likely to receive intravenous therapy until discharge (46 vs. 16%, *p* < 0.001). Receipt of PEIV was associated with a longer mean time to receiving oral antimicrobials (8.7 vs. 3 days, *p* < 0.001). The only factors that impacted EIV days in the multivariable linear regression model were the source of infection (urinary tract) (coefficient −1.54, 95%CI −2.82 to −0.26) and Pitt bacteremia score (coefficient 0.51, 95%CI 0.10 to 0.92). (4) Conclusions: PEIV is common in inpatients with Gram-negative bacteremia. Clinicians should look to avoid PEIV in the inpatient setting.

## 1. Introduction

Excessive intravenous therapy results in multiple negative consequences for patients including duration of hospitalization, costs, and catheter-related bloodstream infections (CRBSIs) [1]. Current guidelines for the management of CRBSIs recommend a duration of therapy of 7–14 days whereas the guideline of antibiotic resistant Gram-negative infections does not offer a recommended duration of therapy [2,3]. Both guidelines are silent on conversion to oral therapy. 

Recent data have suggested that converting to oral therapy produces similar outcomes to intravenous courses [4,5,6]. A large retrospective study further suggested that the transition to oral therapy can occur within the first five days of treatment [7]. However, we do not know from these studies how quickly patients with Gram-negative bacteremia become clinically stable and could be transitioned to oral therapy. This may limit the real-world applicability of using oral antibiotics for Gram-negative bacteremia as many clinicians may be hesitant to promote an early transition to oral antibiotics [8]. 

We hypothesize that many patients with Gram-negative bacteremia experience excessive intravenous therapy. Therefore, we conducted a retrospective cohort study of patients with Gram-negative bacteremia to describe the prevalence of potential excess intravenous treatment and describe the distribution of the length of potential excessive intravenous therapy.

## 2. Materials and Methods

The methods for this retrospective cohort study have been described previously [9]. All adult inpatients at Texas Health Presbyterian Hospital Dallas who had a blood culture positive for *Escherichia coli*, *Pseudomonas aeruginosa*, *Enterobacter* sp., *Klebsiella* sp., *Acinetobacter* sp., or Stenotrophomonas maltophila from 1 January 2008 through 31 August 2011 were included. The study protocol was reviewed and approved by the institutional review boards at both Texas Health Resources (protocol Pro00003313; Approved 1 November 2011). The study was also approved by the Institutional Review Board of Texas Tech University Health Sciences Center (protocol A11-3689; Approved 22 December 2011). First episodes of bacteremia were excluded if they were a mixed or polymicrobial infection. All subsequent episodes of bacteremia were also excluded. Data were manually collected form the study institution’s electronic medical record.

### 2.1. Study Definitions

During the initial study design, the study team agreed upon the definitions to be used. Modified adequate empiric antibiotic therapy was defined as receipt of an antimicrobial for at least the first 24 h that was active against the study pathogen according to the susceptibility report. This modification was made because some patients received only one day of an antimicrobial or the antibiotic was changed after the patient was transferred to a medical floor. Patients were classified as not receiving empiric therapy if they did not receive at least 24 h of an initial empiric antibiotic. Length of stay prior to the positive blood culture being obtained was calculated using the date positive blood culture was obtained minus the date of admission. A patient who had at least one hospital day in the ICU was considered to have an intensive care unit residence as part of their hospital stay. Vasopressor use was considered the utilization of dobutamine, dopamine, isoproterenol, epinephrine, norepinephrine, phenylephrine, vasopressin within 48 h of the obtaining the blood culture that was reported positive. Severity of illness was assessed using the Pitt Bactermia Score [10,11].

Normalization of white blood count (WBC) and temperature were defined for purposes of this study as a patient having a normal WBC count (5.0–11.0 × 10^9^/L) for at least 24 h and remaining afebrile (36.0–38.3 °C) for at least 48 h. Potential excess intravenous therapy was defined as having days of intravenous antibiotic therapy beyond when the both WBCs and temperature had normalized until the patient was discharged or died. 

### 2.2. Statistical Analysis

The primary outcome of this analysis was to describe the frequency and duration of excessive intravenous therapy in patients with Gram-negative bacteremia. Nominal data were analyzed by Chi-squared or Fisher’s exact tests as appropriate. Continuous data were assumed to have a non-normal distribution and were analyzed using a Wilcoxon Rank Sum test. A multivariable linear regression model of factors considered to be reasonably associated with excess intravenous antibiotic therapy was also conducted. Our conceptual model for the multivariable linear regression model was that patients with a urinary source of bacteremia often present from the community and recover rapidly with antibiotics, can be discharged quickly, and therefore have a lower risk of potential excess intravenous therapy. Similarly, we suspected that patients with bacteremia due to *E. coli* or *K. pneumoniae* would be less likely to have multidrug resistance that would meaningfully impact the ability to transition to an oral antibiotic. Pitt bacteremia score was the method we selected to evaluate baseline severity of illness in the model as it provided an ordinal measure of severity for all patients. Alternative measures of severity available within the dataset, such as residing in an intensive care unit or receiving a vasopressor, are nominal characteristics of severity. Days to WBC and temperature normalization were used as a measure as we thought that an extended time to normalization might bias a clinician into leaving a patient on potential excess intravenous therapy. Statistical significance was defined as a two-tailed *p*-value < 0.05. All analyses were performed using STATA 15 (StataCorp. 2011. Stata Statistical Software: Release 15.1., StataCorp LP, College Station, TX, USA).

## 3. Results

Baseline characteristics for the 323 patients included in the cohort are shown in Table 1. Eighty-four patients (26%) did not achieve normalization of WBCs and temperature during hospitalization. As expected, patients who did not achieve normalization of WBCs and temperature had a higher baseline serum creatinine and Pitt bacteremia score. They were also more likely to be in an intensive care unit and/or receiving a vasopressor. Differences in total body weight, infection source, and pathogen were also observed between the groups. Patients who failed to achieve resolution of WBCs and temperature also had significantly higher 30-day mortality (29.8 vs. 7.5%, *p* < 0.001). A lower 30-day mortality rate was observed for patients who had normalization of WBCs and temperature who did not receive any potential excessive intravenous therapy (9.8 vs. 2.7%, *p* = 0.07). Similarly, these patients also had a lower rate of in-hospital mortality (5.5 vs. 0%, *p* = 0.06).

Sixty-nine percent of the 239 patients who achieved normalization of WBCs and temperature during their hospital stay had potential excessive days of intravenous therapy. The baseline characteristics were similar with the exceptions of male sex, length of stay prior to the positive blood culture being obtained, and source of infection. The empiric antibiotic choices for each group are shown in Table 2. Other empiric antibiotics in the no potential excess intravenous group included azithromycin (*n* = 1), aztreonam (*n* = 1), nitrofurantoin (*n* = 1), and sulfamethoxazole (*n* = 1). Other empiric antibiotic for the potential excess intravenous therapy group included azithromycin (*n* = 3), aztreonam (*n* = 5), clindamycin (*n* = 1), nitrofurantoin (*n* = 1), ticarcillin/clavulanic acid (*n* = 1), sulfamethoxazole/trimethoprim (*n* = 2), and tigecycline (*n* = 1). As expected, piperacillin/tazobactam, cephalosporins, and fluoroquinolones were the primary agents received. Approximately one in six patients received combination therapy. The specific combination antibiotic regimens used in the cohort can be found in Appendix A.

The limited antibiotic susceptibility data that were collected in the original dataset are presented in Table 3. All of the included beta-lactams and aminoglycosides had susceptibility rates >90% for the entire cohort. Only 83% of isolates were susceptible to fluoroquinolones. There were no differences in susceptibility rates regardless of whether WBCs and temperature normalized or not. Overall, there were similar findings for patients who had normalized WBCs and temperature regardless of whether they received potential excess intravenous therapy or not. However, levofloxacin susceptibilities were lower for patients who had potential excess intravenous therapy (89 vs. 79%, *p* = 0.05).

Potential excessive intravenous therapy was associated with more days to oral therapy and more days of excessive intravenous therapy (Table 4). Potential excess intravenous therapy was also associated with a longer overall length of stay as well as a longer length of stay after the positive blood culture was obtained. After the removal of 18 patients who achieved normalization of WBCs and temperature, yet died within 30 days of their positive blood culture, patients who received excess intravenous therapy had a higher risk of never changing to oral therapy during their hospitalization (46 vs. 16%, *p* < 0.001). Receipt of potential excessive intravenous therapy was associated with a longer mean time to receipt of oral antimicrobials than patients without potential excess days of intravenous therapy (8.7 vs. 3 days, *p* < 0.001). The duration of potential excessive intravenous antibiotics was 3.7 (95% CI, 2.8–4.6) days.

A urinary tract source of bacteremia was associated with decreased days of excess intravenous therapy in the multivariable linear regression model (Table 5). An increase in Pitt bacteremia score was associated with an increase in excess intravenous days. Pathogen type (presence of *E. coli* or *K. pneumoniae*) and days to normalization of WBCs and temperature were not associated with days of excess intravenous therapy.

## 4. Discussion

This study described how many patients receive potential excess intravenous therapy for Gram-negative bacteremia despite having normalization of WBCs and temperature. We found that 46% of patients who have normalization of these parameters never transition to oral therapy in the hospital. This represents a missed opportunity to potentially decrease their hospital length of stay, decrease their costs, and lessen the risk of CRBSIs.

Tamma and colleagues conducted a large retrospective cohort study to determine if oral step-down therapy within the first five days of therapy was associated with mortality in patients with Enterobacteriaceae bacteremia [7]. The primary source of infection (urinary tract) and pathogens (*E. coli* and *K. pneumoniae*) were also similar to our cohort, but they did not include patients with *P. aeruginosa*. They found 30-day mortality was similar between the groups (13% for both groups, HR 1.03, 95%CI 0.82–1.30) which is similar to our findings. They also found that early oral step-down therapy resulted in a two-day reduction in hospital length of stay which is similar to our reduction in length of stay after the positive blood culture was obtained. Another retrospective study of Enterobacteriaceae-associated bacteremic urinary tract infections (*n* = 241) also found a two day reduction in length of stay with conversion to oral therapy [4].

Our findings add to those of Tamma and colleagues in a few ways. First, we described patients who fail to achieve normalization of their WBCs and temperature during their hospital stay who are likely poor candidates for being transitioned to oral therapy. This is slightly different than the model proposed by Rac and colleagues to provide early prognostic indicators of death at 28 days [12]. Second, we used two commonly collected variables (WBC, temperature) to describe when patients would potentially be eligible for transition to oral therapy rather than a five-day window that requires calculating a Pitt bacteremia score. Three, we documented the number of potential excess intravenous antibiotic days that resulted from not transitioning patients to oral therapy in a timely fashion.

Our study does have limitations including its retrospective single-center design. We acknowledge that the decision to determine that patients are potentially eligible for oral antibiotic therapy based simply on WBC and temperature resolution is incomplete and likely overestimates the amount to potential excess intravenous therapy. However, WBC and temperature normalization are two common criteria that providers do use when evaluating patients for a transition to oral therapy [13,14]. We acknowledge that decisions to switch to oral therapy were dependent on provider decisions that could have been impacted by items not in our dataset. Antibiotic allergies represent an example of this. These allergies, whether real or perceived, may have impacted a patient’s eligibility for oral antibiotics. We did not have this information available to us in the dataset. There are also differences in provider preferences regarding transitioning to oral antibiotics that we could not account for since we did not collect information on providers. 

We did not collect information on other oral medications or dietary status that would have improved our ability to determine if a patient was eligible for oral antimicrobial therapy. Our dataset did not include full susceptibility panels to evaluate whether an oral antibiotic option was available. These limitations likely resulted in an overestimation of patients receiving excessive intravenous therapy. We also acknowledge that resistance patterns may have occurred over the past decade at some institutions that may limit the applicability of our findings. However, recent publications regarding the use of oral antimicrobials for Gram-negative bacteremia speak to the timeliness of this topic in current clinical practice [14,15].

We did not report data on the oral options patients were transitioned to in this paper. This is because several other investigators have previously shown that the degree of bioavailability is not associated with the clinical success of the oral regimen [7,16,17]. We also did not have outpatient data to determine the type and length of antibiotic therapy used as an outpatient. This may have resulted in an underestimation of the number of days of excessive intravenous therapy.

The univariable results presented for the impact of potential excess intravenous therapy on mortality or length of stay were only presented to note that timely transitions to oral therapy did not result in adverse outcomes in our cohort. There are certainly other factors that should be accounted for in a study that is primarily seeking to determine the impact of excess intravenous therapy on mortality and/or length of stay.

## 5. Conclusions

Potential excessive intravenous therapy is common in patients with Gram-negative bacteremia. This conclusion is limited by our retrospective approach and lack of data regarding oral intake of foods or other medications. Since others have demonstrated that oral step-down therapy has similar effectiveness to excess intravenous therapy, we recommend using clinically available data to help ensure the avoidance of excess intravenous days.

## Figures and Tables

**Table 1 pharmacy-09-00133-t001:** Baseline characteristics of the cohort.

Characteristic	White Blood Count and Temperature Did Not Normalize (*n* = 84)	White Blood Count and Temperature Did Normalize (*n* = 239)	*p*-Value	No Potential Excess Intravenous Therapy (*n* = 75)	Potential Excess Intravenous Therapy (*n* = 164)	*p*-Value
Male sex (%)	40.5	37.7	0.65	28.0	42.1	0.04
Race/Ethnicity (%)						
White, non-Hispanic	77.4	73.6		66.7	76.8	
White, Hispanic	6.0	8.0		10.7	6.7	
African American	8.3	11.7	0.67	12.0	11.6	0.30
Other	2.4	0.8		1.3	0.6	
Not reported	6.0	5.9		9.3	4.3	
Age (years)	72 (58.5, 84)	76 (63, 85)	0.36	76 (61, 86)	75.5 (63.5, 85)	0.96
Weight (kilograms)	67 (56, 83.6)	74.8 (63.5, 90.4)	0.003	73.7 (64.3, 93.4)	75.7 (63, 89.4)	0.87
Height (inches)	65 (63.6, 69)	66 (63, 69)	0.99	65 (63, 67.2)	66 (63, 69)	0.15
Cirrhosis (%)	9.5	8.0	0.65	5.3	9.2	0.31
Chronic kidney disease (%)	15.5	22.6	0.17	21.3	23.2	0.75
Cancer (%)	41.7	34.7	0.26	33.3	35.4	0.76
Chronic obstructive pulmonary disease (%)	21.4	13.4	0.08	13.3	13.4	0.99
Diabetes (%)	22.6	33.1	0.07	40.0	29.9	0.12
Baseline SCr (mg/dl)	1.5 (1.1, 2.3)	1.3 (0.9, 2.1)	0.02	1.5 (0.9, 2.4)	1.2 (0.9, 2.0)	0.07
Length of stay prior to positive blood culture (days)	1 (1, 1)	1 (1, 2)	0.30	0 (0, 0)	0 (0, 1)	0.02
ICU residence (%)	32.1	19.3	0.02	16.0	20.7	0.39
Vasopressor use (%)	26.2	14.6	0.02	9.3	17.1	0.12
Pitt bacteremia score	2 (1, 4)	1 (1, 3)	0.001	1 (1, 2)	1 (0, 3)	0.84
Infection source (%)						
Urinary tract	57.1	64.9		81.3	57.3	
Intra-abdominal	11.9	24.3		12.0	29.9	
Intravenous catheter	10.7	3.4	<0.001	1.3	4.3	0.006
Other	10.7	2.9		2.7	3.1	
Undocumented	9.5	4.6		2.7	5.5	
Pathogen (%)						
*Escherichia coli*	61.9	69.0		80.0	64.0	
*Klebsiella pneumoniae*	20.2	17.2		13.3	18.9	
*Pseudomonas aeruginosa*	14.3	7.5	0.05	4.0	9.2	0.13
*Enterobacter* sp.	1.2	5.9		2.7	7.3	
Other	2.4	0.4		0.0	0.6	
Adequate empiric antimicrobials (%)	89.2	94.5	0.10	98.7	92.6	0.07

**Table 2 pharmacy-09-00133-t002:** Empiric therapy choices.

Agent (%)	No Potential Excess Intravenous Therapy (*n* = 75)	Potential Excess Intravenous therapy (*n* = 164)	*p*-Value
Aminoglycosides	2.7	2.4	1.00
Fluoroquinolone	22.7	21.3	0.82
Piperacillin/tazobactam	37.3	28.7	0.18
Cephalosporin	30.7	28.7	0.75
Carbapenem	10.7	19.5	0.09
Other agent	5.3	7.9	0.47
Combination therapy	16	17.7	0.75

**Table 3 pharmacy-09-00133-t003:** Limited antibiotic susceptibility data.

Agent (Percent Susceptible %)	White Blood Count and Temperature Did Not Normalize (*n* = 84)	White Blood Count and Temperature Did Normalize (*n* = 239)	*p*-Value	No Potential Excess Intravenous Therapy (*n* = 75)	Potential Excess Intravenous Therapy (*n* = 164)	*p*-Value
Amikacin	97.6	99.6	0.16	100	99.4	1.00
Cefepime	95.2	96.7	0.51	97.3	96.3	1.00
Gentamicin	91.6	92.5	0.79	94.7	91.5	0.44
Levofloxacin	86.8	82.0	0.32	89.3	78.7	0.05
Meropenem	97.6	98.7	0.61	98.7	98.8	1.00
Piperacillin–tazobactam	92.8	94.6	0.55	96.0	93.9	0.76

**Table 4 pharmacy-09-00133-t004:** Outcomes of patients with and without potential excess intravenous therapy presented as median and interquartile range.

Outcome	No Potential Excess Intravenous Therapy (*n* = 75)	Potential Excess Intravenous Therapy (*n* = 164)	*p*-Value
Days to oral antimicriobial therapy	3 (2, 4)	6 (4, 11)	<0.001
Days of excess intravenous antimicrobial therapy	0 (0, 0)	2 (1, 4)	<0.001
Length of stay after positive blood culture (days)	5 (3, 8)	7 (4.5, 11)	<0.001
Total length of stay (days)	5 (3, 8)	8 (5, 13)	<0.001
Death within hospitalization	0	5.5	0.06
Death at 30 days	2.7	9.8	0.07

**Table 5 pharmacy-09-00133-t005:** Linear regression analysis of potential excess intravenous days.

Characteristic	Coefficient	95% CI
Days to normalization of WBCs and temperature	−0.15	−0.35 to 0.05
Pathogen: *E. coli* or *K. Pneumoniae*	−1.10	−2.88 to 0.68
Pitt Bacteremia Score (baseline)	0.51	0.10 to 0.92
Source: UTI	−1.54	−2.82 to −0.26

## Data Availability

The data that support the findings of this study are available from the corresponding author, RGH, upon reasonable request.

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
