# Peer review of "Potential Excess Intravenous Antibiotic Therapy in the Setting of Gram-Negative Bacteremia"

_pharmacy, 2021, doi:10.3390/pharmacy9030133_

Round 1
Reviewer 1 Report
I have reviewed this retrospective cohort study examining excessive intravenous therapy in patients with gram-negative bloodstream infection (BSI). The study hypothesis is interesting. However, there are several limitations that influence the study results and conclusions.
- The definition of clinical stability seems arbitrary. Resolution of fever is not an indicator of prognosis or the decision to switch from IV to oral therapy. Most clinicians decide to switch to oral therapy in hemodynamically stable patients who are tolerating diet. Prognostic early clinical failure criteria in patients with gram-negative BSI are already established in the literature.
- The authors examine risk factors for excessive intravenous therapy. However, the authors do not present the full results of the univariate and multivariate models.
- Second, the model lacks few of the most important variables such as lack of reliable oral options (based on susceptibility results or allergies) and presence of complications or risk factors for complicated BSI as prosthetic cardiac devices or joints.
- The results imply that excessive intravenous therapy leads to longer hospital length of stay and higher mortality. I assume higher Pitt bacteremia score and deeper source of infection are the reason for longer hospital length of stay and higher mortality in patients receiving excessive intravenous therapy.
- The study conclusions do not match the results. I don’t believe that the authors have established criteria for transitioning patients from IV to oral therapy. Those criteria were made arbitrarily without thorough review of the literature.
- The study period goes back to over a decade ago. Clinical practice and antimicrobial resistance rates have changed since then.
- Did the study include consecutive cases of BSI due to any gram-negative bacteria or just the listed select pathogens?
- It is essential to report susceptibility results to various oral antibiotics and the oral agents used in the study. The results are incomplete without these data.
Reviewer 2 Report
The manuscript entitled "Excess intravenous antibiotic therapy in the setting of gram- negative bacteremia" describes efficacy of excessive intravenous antibiotic therapy.
Comments.
1) What is known about the resistance pattern of causative agents isolated from bloodstream infections?
2) In Table 1 at the pathogens part you indicated "other". Could you please, add all pathogens to this table?
3) In Table 2 at the antibiotics, you indicated "other". Could you please add all agents to this table?
4) In Table 2 stands "Combination therapy". Could you please indicate in the table or in the text the applied combinations?
Reviewer 3 Report
The manuscript titled “Excess intravenous antibiotic therapy in the setting of gram-negative bacteremia” by Ashley R. Selby et al. I have reviewed it is an interesting topic, for which there are still no clear and defined guidelines. In my opinion the authors have yet to clarify some aspects of their study
- The introduction is very poor and must be improved.
- As regards the Materials and Methods section, the Authors should give much more detailed information such as, for example, which hospital(s) have been considered, from which database they have collected the reported data, the antibiotic therapy and the specific dosage that have been administered to patients…
Finally, the Authors must entirely rewrite their manuscript according to the template provided by Pharmacy, respecting the instructions for the authors. Also the references list should be modified accordingly. According to Pharmacy instructions, the 200 words abstract should have a very different organization such as Background, Methods, Results and Conclusion.
Round 2
Reviewer 1 Report
I thank the authors for revising this manuscript based on reviewers’ comments to make it more informative for the readers. I still have few suggestions for improvement.
1- Intra-abdominal source of infection seems to be associated with potential excessive intravenous therapy based on Table 1 results (57% vs. 30%). This is clinically conceivable since many patients with intra-abdominal infections may not be able to take oral therapy that early in the course of illness regardless of temperature or leukocytosis. Intra-abdominal source of infection should be included in the linear regression model for risk factors of potentially excessive intravenous therapy. The Discussion section should be expanded to comment on this variable.
2- Levofloxacin resistance also seems to be associated with potential excessive intravenous therapy based on Table 3 results (p=0.05). Levofloxacin susceptibility was 10% lower in patients who received potentially excessive intravenous therapy than comparators. This is also clinically conceivable since fluoroquinolone resistance limits reliable oral antibiotic options for switch therapy. Fluoroquinolone resistance should be added to the linear regression model for risk factors of potentially excessive intravenous therapy. The Discussion section should be revised to emphasize the association between fluoroquinolone resistance and potentially excessive intravenous therapy.
3- Table 4 title should be changed to “outcome of patients with and without potential excessive intravenous therapy”. The current title of Table 4 implies that these are consequences of excessive intravenous therapy.
Author Response
Please find our attached response.
